# A Comprehensive Assessment of Qualitative and Quantitative Prodromal Parkinsonian Features in Carriers of Gaucher Disease—Identifying Those at the Greatest Risk

**DOI:** 10.3390/ijms232012211

**Published:** 2022-10-13

**Authors:** Michal Becker-Cohen, Ari Zimran, Tama Dinur, Maayan Tiomkin, Claudia Cozma, Arndt Rolfs, David Arkadir, Elena Shulman, Orly Manor, Ora Paltiel, Gilad Yahalom, Daniela Berg, Shoshana Revel-Vilk

**Affiliations:** 1Gaucher Unit, Shaare Zedek Medical Center, Jerusalem 9103102, Israel; 2Braun School of Public Health and Community Medicine, Hebrew University of Jerusalem, Jerusalem 9124001, Israel; 3Faculty of Medicine, Hebrew University of Jerusalem, Jerusalem 9124001, Israel; 4Centogene GmbH, 18055 Rostock, Germany; 5Medical Faculty, University of Rostock, 18051 Rostock, Germany; 6Arcensus GmbH, 18055 Rostock, Germany; 7Department of Neurology, Hadassah Medical Center, Hebrew University of Jerusalem, Jerusalem 9124001, Israel; 8Department of Neurology, Shaare Zedek Medical Center, Jerusalem 9103102, Israel; 9Department of Neurology, Christian-Albrechts-University of Kiel, 24118 Kiel, Germany

**Keywords:** Parkinson’s disease, Gaucher disease carriers, carriers of GBA1 variants, prodromal Parkinson disease

## Abstract

Carriers of GBA1 gene variants have a significant risk of developing Parkinson’s disease (PD). A cohort study of GBA carriers between 40–75 years of age was initiated to study the presence of prodromal PD features. Participants underwent non-invasive tests to assess different domains of PD. Ninety-eight unrelated GBA carriers were enrolled (43 males) at a median age (range) of 51 (40–74) years; 71 carried the N370S variant (c.1226A > G) and 25 had a positive family history of PD. The Montreal Cognitive Assessment (MoCA) was the most frequently abnormal (23.7%, 95% CI 15.7–33.4%), followed by the ultrasound hyperechogenicity (22%, 95% CI 14–32%), Unified Parkinson’s Disease Rating Scale part III (UPDRS-III) (17.2%, 95% CI 10.2–26.4%), smell assessment (12.4%, 95% CI 6.6–20.6%) and abnormalities in sleep questionnaires (11%, 95% CI 5.7–19.4%). Significant correlations were found between tests from different domains. To define the risk for PD, we assessed the bottom 10th percentile of each prodromal test, defining this level as “abnormal”. Then we calculated the percentage of “abnormal” tests for each subject; the median (range) was 4.55 (0–43.5%). Twenty-two subjects had more than 15% “abnormal” tests. The limitations of the study included ascertainment bias of individuals with GBA-related PD in relatives, some incomplete data due to technical issues, and a lack of well-characterized normal value ranges in some tests. We plan to enroll additional participants and conduct longitudinal follow-up assessments to build a model for identifying individuals at risk for PD and investigate interventions aiming to delay the onset or perhaps to prevent full-blown PD.

## 1. Introduction

The first observation of an association between Gaucher disease (GD) and Parkinson’s disease (PD) was reported by our group already in 1996 [1]. The first six patients were characterized by relatively mild GD and severe PD, with an early age of onset, poor treatment response, and severe cognitive decline. A few years later, it became apparent that beyond the association between GD and PD, carriers of GD (carriers of a variant in the GBA1 gene) are also at an increased risk for developing PD [2]. Not unexpectedly, a high prevalence (10–25%) of GBA1 variants was identified in PD patients [3,4]. In a meta-analysis of patients with PD comparing the different variants in the GBA1 gene, it was found that carriers of the N370S variant (new nomenclature c.1226A > G; p.N409S) had a 3-fold risk for developing PD, whereas carriers of other GBA1 variants exhibited a 10–15-fold risk for developing PD compared to the general population [5]. As in our original mini-series from 1996, GBA-associated PD was also associated with an earlier age of onset, a faster deterioration of motor functions, and a higher frequency and faster progression of cognitive decline than non-GBA-associated PD, although considerable clinical variation was seen [6].

Before the onset of the typical motor features of PD, there is a period of between 10 and 20 years in which patients develop a series of different signs and symptoms, known today as prodromal PD [7]. These are mainly non-motor symptoms such as sleep disorders, depression, smell loss, and constipation. Although well recognized, testing for prodromal PD has not entered routine practice, neither by primary care physicians nor by movement disorder specialists. It may be argued that early recognition of risk without beneficial therapy will only provoke depression and anxiety [8]. This is even more relevant insofar as many individuals with signs and symptoms of prodromal PD may never develop the clinical disease. In this context, it should be appreciated that there is a lack of standardized methodology to assess prodromal PD. This flaw is increasingly encountered by the International Parkinson and Movement Disorder Society (MDS) research criteria on prodromal PD [8] and their updated version [9]. In the criteria, the likelihood is calculated according to sex, age, family history of PD, known PD-related genetic variants, environmental factors (exposure to pesticides and solvents, caffeine consumption, and smoking habits), erectile and urinary dysfunction, and tests for motor and non-motor clinical symptoms [8,9].

The characteristic motor symptoms of PD become evident when dopaminergic neurons of the substantia nigra (SN) are damaged or destroyed, causing dopamine levels to drop and a disruption of normal signaling. Consensus opinion asserts that at least 60% of neuron destruction is required for motor symptoms to become evident. Thus, symptoms appear at a late stage, minimizing the chance to salvage functioning neurons [10,11,12]. Therefore, there is a rationale for identifying PD at the prodromal stage before motor symptoms appear and evaluating treatments to prevent or slow PD progression. Currently, PD is a relentlessly progressive disorder with only symptomatic therapeutic options. The progressive motor impairment associated with PD means that affected patients eventually require professional and/or caregiver support. The quality of life of PD patients is often adversely affected by both the motor and non-motor symptoms of the disease, including depression, anxiety, apathy, sleep disturbances, and cognitive impairment [13].

The Shaare Zedek Medical Center (SZMC) dedicated center for GD in Israel is the largest Gaucher Unit worldwide, with over 900 patients managed over 30 years. This is partly explained by the higher prevalence of GD among the Ashkenazi Jewish population [14]. Thus, we have access to a large cohort of obligatory carriers of a variant in the GBA1 gene (GBA carriers), enabling us to study prodromal PD using non-invasive tests in an unselected high-risk population. In 2017 we launched a study on GBA carriers to ascertain the prevalence of prodromal PD features, assessing them longitudinally and thus finally identifying candidates for a PD prevention trial. Herein, we report on the first cohort.

## 2. Results

### 2.1. Study Cohort

The characteristics of the first 98 consecutive GBA carriers who completed at least 50% are shown in Table 1. The most common *GBA1* variant, found in two-thirds of the participants, was N370S, followed by the 84GG, found in more than 10% of participants. A quarter of the participants reported a family history of PD in 1st and/or 2nd-degree relatives. Having the milder *GBA1* variant, i.e., N370S, was not associated with age, sex, or family history of PD.

### 2.2. Abnormal Prodromal Tests

Over 20% of the cohort demonstrated abnormally elevated SN hyperechogenicity and an abnormally low total MoCA score (Table 2). More than 10% of the cohort had the following abnormalities: UPSIT smell test, urinary and erectile dysfunction, visual-spatial perception in the NeuroTrax cognitive test, REM sleep behavior, Epworth sleepiness, and UPDRS-III (Table 2).

### 2.3. Variables Associated with Abnormal Prodromal Tests

Male subjects had higher TCS measurements, higher scores in two NeuroTrax subsets (visual-spatial and motor skills), and lower Beck depression inventory scores compared to females (Table 3 (a)). Age was associated with scores on color, smell, and motor skills (Perdue pegboard, UPDRS III and iTUG) (Figure 1). Subjects with a family history of PD had lower scores in two NeuroTrax subsets (executive function, visual-spatial) and in the NeuroTrax global cognitive score (Table 3 (b)). The type of *GBA1* variant (mild vs. severe) was not associated with prodromal features. 

### 2.4. Correlations

A correlation matrix was constructed to assess correlations among the different prodromal features (Figure 1). Strong correlations were found within the NeuroTrax tests and within the Perdue pegboard tasks. Moderate correlations were seen between the NeuroTrax and color, MoCA, and the different motor tests. Poor correlation was seen between the two sleep assessment tests (r = 0.13).

### 2.5. Most “Abnormal” Prodromal Tests

The median (range) percentage of “abnormal” tests per subject was 4.55 (0–43.5%), with different median percentages according to the *GBA1* variants (Table 4).

## 3. Discussion

In this study, which is to the best of our knowledge, the largest single-center study of its kind to assess non-PD GBA carriers [15,16], we selected a wide range of tests covering all clinical domains of prodromal PD. All the performed tests were non-invasive, measurable, and user-friendly, as our evaluated subjects are practically healthy. Some of the tests included in the current study, such as color discrimination, NeuroTrax computerized battery, and iTUG mobile platform, were not previously reported in non-PD GBA carriers. The most common abnormalities were sleep and smell assessment, UPDRS-III, TCS hyperechogenicity, and MoCA. Although a high rate of sleep abnormalities [17], high TCS scores [18,19], and lower MoCA scores [16] have been previously reported in cohorts of GBA carriers, these findings were identified in different tests. Our study is the first to show this in a single study, enabling us to count the total number of abnormalities for each subject and identify possible candidates for future PD-prevention studies.

We analyzed the results of the prodromal features according to the different PD risk factors, including sex, age, family history, and genotype. The poorer performance in visual-spatial perception and motor skills and the higher depression scores seen in women compared to men was similar to the general population [20,21,22]. The higher hyperechogenicity found in men compared to women has previously been reported by three different groups in China [23,24,25]. Whether it is correlated with the higher rate of PD found in males or an actual difference between genders should be further studied.

Advanced age impacts the performance in many of the prodromal tests; however, only two tests, i.e., color discrimination and NeuroTrax, include an adjustment for age. Among the tests without age adjustment, we noticed a weak correlation between age with motor skills, MoCA, and smell. Although TCS was shown to be positively correlated to age in healthy controls [25], it seems that this correlation does not exist with PD patients [26]. This suggests that the TCS score may be a stable marker in patients with PD throughout the course of the disease [12].

The strong association of positive family history with poorer executive function and visual-spatial scores (NeuroTrax), although still within the normal range, is interesting and may reflect selection bias, i.e., subjects with a family history of PD and cognitive changes were more inclined to enroll to this screening study. However, to avoid bias, in cases when more than one family member participated in this study, we only included the older person in this report.

Unexpectedly and in contrast to the literature indicating a greater risk for PD and a more progressive and severe course of PD among carriers with non-N370S variants versus N370S [27,28,29], our results did not reveal significant differences in the prodromal findings between subjects with N370S and non-N370S. This discrepancy may be related to an ascertainment bias of carriers with a family history of PD, having a higher incentive to volunteer for the study, or simply since the non-N370S variants are under-represented. Nevertheless, subjects with the L444P variant exhibited the highest percentage of “abnormal” tests (23%) compared to 4.76% in subjects carrying the N370S variant. This difference was not statistically significant, probably because of the small number of L444P variants in our cohort. A larger cohort of subjects could result in different conclusions.

The use of TCS has gained greater acceptance in recent years despite earlier concerns that it is inferior to FDOPA PET-CT and DatSCAN SPECT. A recent report from the Sidransky group at the NIH confirms the value of this imaging modality in assessing nigral echogenicity compared to striatal presynaptic dopamine synthesis capacity as observed in [18 F]-FDOPA PET-CT [30]. However, in that paper, hyperechogenic findings were documented only among subjects who had already developed PD, and accordingly raised questions about their utility as predictive tools in at-risk subjects. In our study, enlarged hyperechogenic areas were found in 18.5% of GBA carriers without PD. Moreover, previous research from our group has demonstrated similar findings (17.96%) among both GBA carriers and patients with GD without PD [18]. Taken together, our previous findings support the use of TCS in assessing prodromal PD features. We have also seen anecdotal cases where worsening in the hyperechogenicity occurred in parallel with the actual development of early motor signs of PD (unpublished). Two possible explanations for the discrepancy between the findings of the two groups could be the larger group of GBA carriers available for our study and the fact that the TCS is an operator-dependent test.

When surveying the battery of available tests, the broad diversity of the findings should improve our ability to pinpoint a sub-group of those tests that will be used in future longitudinal studies, as well as to select candidate subjects for future interventional studies targeting PD prevention to those at the highest risk [31]. These will be possible particularly because most tests are quantitative (not just qualitative) and can be objectively measured. This may even be more important given the observation that the prodromal period, among GBA carriers who developed PD, was shorter and more severe [32]. Our study’s ultimate goal has been to identify those GBA carriers at the most advanced prodromal stage, that would justify future recruitment into an interventional clinical trial. Selecting candidates for an interventional preventive study is challenging, particularly when no similar clinical trial for PD prevention has ever occurred. We searched for subjects with test scores in the worst 10th percentile (“abnormal”) and found that around 20% of subjects had more than 15% “abnormal” tests. It should be noted that we did not use the updated MDS research criteria for prodromal PD for this report in part because some of the tests we used are not included in the calculator. By increasing the number of healthy controls and patients with PD we should be in a position to estimate the likelihood ratio (LR) and calculate the probability of PD.

One of the limitations of the study is an ascertainment bias of those individuals who have experienced the agony of GBA-related PD in first-degree relatives, possibly contributing to the 25% family history of PD included in the list of risk factors. However, a similar number was found more than a decade ago when we performed a paper-based questionnaire to assess the rate of PD among individuals with GD compared to the rate in their spouses. Among the GD patients, 27.3% reported having a relative with PD compared to 12.3% in the control group, which was statistically significant (*p* = 0.05) [33]. Nevertheless, this limitation is only applicable if one wants to assess the prevalence of various PD prodromal features among the entire population of GBA carriers. Our main goal in this study was primarily to identify those patients who are at the greatest risk.

Additional limitations include some incomplete data for specific tests due to technical issues (lack of bone window for assessing SN hyperechogenicity, temporary lack of access to OCT measurements, technical problems with iTUG application, etc.), the lack of well-characterized normal value ranges for some of the tests and the absence of a non-GBA control group.

## 4. Materials and Methods

### 4.1. Study Cohort

Obligatory GBA carriers between the ages of 40 and 75 were approached to participate in this study. They represented parents, siblings, and children of patients with GD and known GBA carriers who sought pre-marital, preconception, or prenatal genetic counseling in the past. Only subjects confirmed by whole gene sequencing to be carriers of a *GBA1* variant were included in this investigative sample. Patients with GD, known PD, other neurodegenerative or severe illnesses, and normal *GBA1* sequencing were excluded. In the case of siblings, only the older sibling was included in the analysis. The institutional review board (IRB) of SZMC approved the study design (NCT05253560), and all subjects provided written informed consent.

### 4.2. Prodromal Tests

Participants were administered pre-defined, non-invasive tests for prodromal PD and assessment of risk factors. Two study coordinators (MBC, TD) administered all testing and each oversaw the same group of tests. Transcranial sonography (TCS) was performed by a single imaging technician (MT).

Participants were questioned regarding risk factors for PD including age, sex, type of *GBA1* variant, coffee and tea consumption, smoking habits, exposure to solvents and pesticides, and family history of PD (1st and/or 2nd-degree relatives with PD). The type of *GBA1* variant was defined as mild (N370S (c.1226A > G)) vs. severe (all other). A caffeine-free diet was defined as <0.5 cup/day. The prodromal tests aimed to assess different aspects of PD were divided into five domains: imaging, sensory and autonomic, cognitive and mental, sleeping disorder, and motor evaluation.

### 4.3. Imaging

Transcranial sonography (TCS) is a useful non-invasive diagnostic tool to detect hyperechogenicity of the SN area in the brain. The hyperechogenicity is caused at least in part by increased amounts of iron, which is known to drive PD pathology [34]. The echogenicity of the SN was tested by Philips/HP Sonos 5500 ultrasound, S4/2.0–2.5. A cut-off value ≥ 0.2 cm^2^, found in more than 90% of patients with PD, was considered abnormal (sensitivity 90.7%, specificity 82.4% and positive predictive value 92.9%) [35,36]. Extreme hyperechogenicity was defined when TCS > 0.27 cm^2^.

### 4.4. Sensory and Autonomic Assessments

Color discrimination deficit is a common nonmotor manifestation of PD, partly explained by losing dopaminergic neurons in the retina [37,38,39]. Color discrimination was analyzed using Farnsworth-Munsell 100 Hue Color Vision Test. The test includes four boxes with a fixed number of color shade caps with slight shade differences that needed to be ordered correctly. The result of each box was entered into a computerized system and automated. The 95% CI normal for age of the total error score (TES) was used for analysis [40].

Hyposmia occurs in ~90% of early-stage cases of PD [41]. The shorter version of The University of Pennsylvania Smell Identification Test (UPSIT) smell test was used to evaluate hyposmia using twelve cards, each with a different smell (Brief Smell Identification Test™ (B-SIT^®^)). A score of less than eight was considered abnormal, 8–10 indeterminate, and 11 or more were deemed normal [42,43].

Orthostatic hypotension (OH), defined as a fall of at least 20 mmHg systolic or at least 10 mmHg diastolic blood pressure after 3 min of active standing or head-up tilt, is a key manifestation of autonomic dysfunction in PD. In a meta-analysis, the pooled odds ratio of OH with PD was 4.343 (95% CI 3.323–5.676) with low heterogeneity [44].

Subjects were asked about their weekly bowel movement frequency; a frequency of ≤0.5 per day was rated as abnormal (constipation) [45]. Urinary and erectile dysfunction were assessed according to the MDS criteria [9].

### 4.5. Cognitive and Mental

The Montreal Cognitive Assessment (MoCA) is a brief cognitive screening tool to assess cognitive impairment. A score of 26 and above was considered normal (25 and above for those with only a high school education). In newly diagnosed, untreated patients with PD, a MoCA score of <26 was found in 22% of patients and 1% met criteria for dementia-level impairment (i.e., MoCA score < 21) [46]. For the visuospatial/executive section, a score of 3 and below was deemed to be abnormal, and for language fluency, a score of 0 was deemed abnormal.

The NeuroTrax computerized battery (www.neurotrax.com (accessed on 31 July 2022)) includes seven sections: memory, executive function, attention, information processing speed, visual-spatial, verbal function, and motor skills [47]. A score of 100 and above was considered normal, and a score of less than 85th percentile (-SD) was abnormal. The computerized test score was adjusted to age and level of education.

The Beck Depression Inventory self-administered questionnaire, with 21 included questions was used to screen, diagnose, and measure the severity of depression. A cutoff of 14 and above indicated depression [48,49]. The frontal assessment battery test (FAB) evaluates executive dysfunction and, for this, a score of less than 16 was considered abnormal [50].

### 4.6. Sleep Disorder

Individuals were screened for rapid eye movement (REM) sleep behavior and daytime sleepiness. Idiopathic REM sleep behavior disorder is an important risk factor in the diagnosis of PD [39]. The REM questionnaire was completed by the participant. A score of five or above was regarded as abnormal [51]. The Epworth sleepiness scale (ESS) for assessing daytime sleepiness includes an eight-question questionnaire scoring between 0 and 24; the higher the score, the higher the chances of dozing off [52,53]. A cutoff above ten was considered pathological sleepiness.

### 4.7. Motor

The Perdue pegboard test (PPT) measures the movement of fingers, hands, and arms in four different tasks [54]; placing as many pins as possible down on the row within 30 s of the right hand, left hand, and both hands and using both hands simultaneously while assembling pins, washers, and collars within 60 s (assembly) [55]. Each task is performed three times and the average is calculated. Average placement of <11 pegs at 30 s using both hands was considered abnormal ([56]; and Postuma personal communication).

The Unified Parkinson’s Disease Rating Scale (UPDRS) part III measures gross motor function [57]. A score above 6, excluding postural and action tremors, was considered abnormal [45].

The Timed Up and Go (iTUG) (EncepaLog^TM^) is a platform that utilizes smartphone internal motion sensors for conducting motor evaluation [58,59] including general walking score, step-to-step persistency, hand sway, and rotation time. For this study, the average time (3 rounds) that took the subject to get up from sitting, walk 10 m, turn, walk back and sit down was calculated. The range of normalcy for this test was unavailable.

### 4.8. Genetic Analysis and Biobank

Blood samples were taken from each participant; blood was applied onto a filter card and sent to Centogene, Rostock, Germany for *GBA1* sequencing and lyso-Gb1 levels. Future sequencing of a panel of 30 genes related to PD. Other blood tubes were centrifuged and frozen as red blood cells, serum, and plasma for future analysis.

### 4.9. Statistical Analysis

To report summary descriptive statistics, we used median (range) for continuous variables. The Shapiro-Wilk normality test, used to test for normality, showed a skewed distribution of the variables. For nominal data, we reported absolute and relative frequencies. The Chi-square test was used to evaluate associations between categorical variables and abnormal prodromal testing, such as sex, family history of PD, and type of *GBA1* variant. The Mann-Whitney test was used to compare the results of two independent non-parametric variables. The non-parametric Spearman correlation was used to study the correlation between age and the results of prodromal tests and the correlation within the prodromal tests. For the correlation analysis, we lined all prodromal features on a scale from worst to best. Thus, parameters in the opposite direction, i.e., from best to worst, were transformed into negative numbers. Correlation strengths were defined as very strong > 0.75, strong 0.5–0.75, moderate 0.25–0.5 and weak < 0.25. To define a group that may be at higher risk for developing PD, for each test (lined on a scale from worst to best), the bottom 10 percent of the distribution was defined as “abnormal”. Then we calculated for each subject the percentage of “abnormal” tests from the total performed. Statistical analysis was performed using SPSS version 26.0 and R programming. Due to multiple testing, a two-sided significance level of α < 0.01 was considered statistically significant.

## 5. Conclusions

Our current plan is to compare the findings with an unselected non-GBA control group, to continue screening GBA carriers over the age of 40 years to allow an even more robust database. This would allow us to perform a longitudinal follow-up of those who have been tested for more than two years previously, and to screen those individuals with the greatest PD risk for our pending interventional clinical trial. It is possible that advanced prodromal features of PD are manifestations of PD in evolution. Candidate therapeutic options include GBA-specific pharmacological chaperons and anti-inflammatory drugs such as colchicine [60] or ursolic acid which have demonstrated impact on neurodegeneration in various animal models including the MPTP-Induced intoxicated mouse model [61] Showing reversibility or even stability relative to a placebo control group could be a groundbreaking discovery on the road to PD prevention [62].

## Figures and Tables

**Figure 1 ijms-23-12211-f001:**
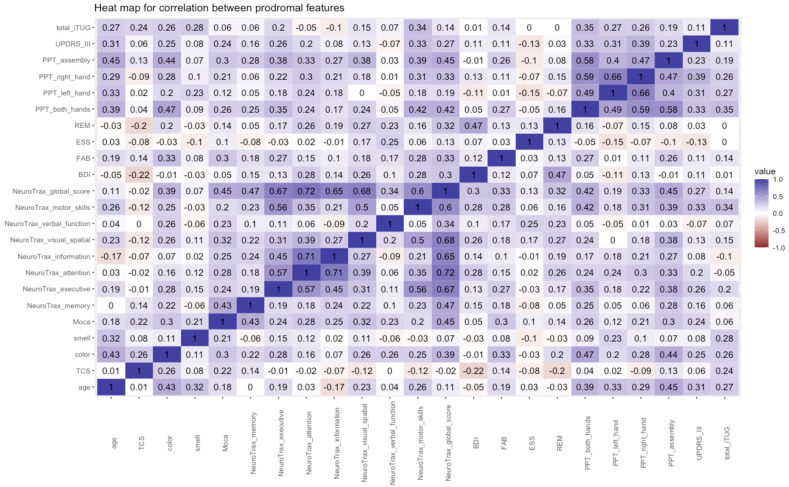
Correlation analysis of prodromal tests lined on a scale from worse to best. Correlation strengths were defined as strong > 0.75, moderate 0.5−0.75, weak 0.25−0.5 and no relationship < 0.25. TCS, transcranial cranial sonography; BDI, Beck depression innovatory; FAB, frontal assessment battery; ESS, Epworth sleepiness scale; REM, rapid eye movement, PPT, Perdue pegboard test; iTUG, Timed Up and Go.

**Table 1 ijms-23-12211-t001:** Characteristics of the 98 GBA carriers enrolled in the study.

	Total
**Male**	43
**Age, median (range)**	51 (40–73)
**Age categories**	
40–44	22
45–49	20
50–54	16
55–59	11
60–64	11
65–69	11
70–74	7
** *GBA1* ** **variant ***	
N370	71
84GG	12
L444P	5
R496H	4
V394L	3
Other **	3
**Relatives with Parkinson’s disease**	25
1st degree	15
2nd degree	13
**Caffeine use *****	82
**Smoking *****	
Current	9
Former	23
Never	61
**Regular pesticide exposure *****	6
**Occupational solvent exposure *****	4

* Old nomenclature. ** other variants- M85T, p330*, T410M. *** answered only by 93 participants.

**Table 2 ijms-23-12211-t002:** Median (range) and rate of abnormal prodromal tests in the five domains.

	Median (Range)	Abnormal Cutoff *	Abnormal/Tested (%)
*(a) Imaging*			
**Transcranial sonography, cm^2^**	0.12 (0–0.28)	≥0.2	20/91 (22)
*(b) Sensory and autonomic*			
**Color discrimination test** **(Total error score)**	41.5 (4–171)	Age 40–49 years > 100 Age 50–59 years > 130 Age 60–69 years > 170 Age 70–79 years > 195	2/92 (2.2)
**UPSIT smell test**	10 (0–12)	<8	12/97 (12.4)
**Orthostatic hypotension**	NR	>20 SBP or > 10 mmHg DBP	6/98 (6.1)
**Bowel movement (daily)**	NR	≤0.5	4/98 (4.1)
**Urinary dysfunction**	NR	Yes/No	15/92 (16.3)
**Erectile dysfunction**	NR	Yes/No	7/43 male (16.6)
*(c) Cognitive and mental*			
**Beck depression inventory**	4 (0–25)	≥14	9/96 (9.4)
**Frontal assessment battery**	18 (15–18)	<16	2/96 (2.1)
**MoCA-** Total score Visuospatial/executive Language fluency	28 (22–30) 4 (2–5) 1 (0–1)	≤25 ≤3 =0	23/97 (23.7) 26/98 (26.5) 15/98 (15.3)
**NeuroTrax-** Memory Executive function Attention Information processing Visual-spatial Verbal function Motor skills Global cognitive score	104.5 (61.1–114.5) 106.1 (77.7–134.2) 103.9 (63.5–119.4) 101.7 (69.1–139) 109.1 (59–132.3) 107.6 (38.2–116.4) 108.4 (69.2–120.5) 104.7 (89.4–120.8)	<85 percentiles <85 percentiles <85 percentiles <85 percentiles <85 percentiles <85 percentiles <85 percentiles <85 percentiles	8/98 (8.1) 3/98 (3.1) 3/98 (3.1) 7/97 (7.2) 10/98 (10.2) 6/98 (6.1) 1/97 (1.03) 0/98 (0)
*(d) Sleeping disorder*			
**REM sleep behavior disorder**	2 (0–12)	≥5	11/96 (11.5)
**Epworth sleepiness scale**	6 (0–20)	>10	11/98 (11.2)
*(e) Motor*			
**Perdue pegboard**	14.3 (8.2–18.8)	<11 **	2/96 (2.1)
**UPDRS-III**	2 (0–13)	>6 ***	16/93 (17.2)
**iTUG time, seconds**	19.9 (14.6–25.9)	NA	NA

* Based on literature review (as detailed in the methods). ** Based on placing as many pins as possible down on the row with both hands at 30 s (average of three times). *** Excluding postural and action tremor. UPSIT, University of Pennsylvania Smell Identification Test; NR, non-relevant; SBP, systolic blood pressure; DBP, diastolic blood pressure; MoCA, Montreal Cognitive Assessment; REM, rapid eye movement; UPDRS-III, Unified Parkinson’s Disease Rating Scale part III; iTUG, Timed Up and Go; NA, non-available.

**Table 3 ijms-23-12211-t003:** Associations of sex and family history of Parkinson’s disease with abnormal prodromal tests *.

*(a) Sex*	*Female (n = 55)*	*Male (n = 43)*	*p*
**Transcranial sonography, cm**	0.09 (0–0.27)	0.13 (0.06–0.28)	<0.001
**NeuroTrax** Visual-spatial Motor skills	105.1 (64.6–123.9) 106 (69.2–119.5)	113.6 (89.1–132.4) 111.1 (93.7–120.5)	0.003 <0.001
**Beck depression inventory**	5 (0–25)	2 (0–20)	<0.001
** *(b) Family history* **	* **No (n = 73)** *	* **Yes (n = 25)** *	
**NeuroTrax** Executive function Visual-spatial Global cognitive score	109.8 (79.7–134.2) 110.2 (64.6–132.4) 106.1 (89.9–120.5)	100.8 (77.7–122.9) 99.8 (69.3–120.5) 101.7 (89.4–112.4)	0.003 0.008 0.004

Data are shown as median (range) * Only significant associations are shown.

**Table 4 ijms-23-12211-t004:** Percentage of “abnormal” tests according to *GBA1* variant.

*GBA1* Variant	Number of Subjects	% of “Abnormal” Tests per Subject *
N370	71	4.76 (0–43.48)
84GG	12	7.05 (0–34.78)
L444P	5	23.81 (0–34.78)
R496H	4	8.70 (0–31.82)
V394L	3	0.00 (0–4.35)
Other **	3	4.35 (0–4.35)

* Median (range). ** other variants: M85T, p330*, T410M.

## Data Availability

All data are kept at the Gaucher unit according to GCP and institution regulations.

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
