# Peer review of "A Comprehensive Assessment of Qualitative and Quantitative Prodromal Parkinsonian Features in Carriers of Gaucher Disease—Identifying Those at the Greatest Risk"

_ijms, 2022, doi:10.3390/ijms232012211_

Round 1

Reviewer 1 Report

I have some comments on the manuscript. 

1. Write limitations of your study at the end of abstract section. 

2. Elaborate your major objectives and approaches in the last paragraph of introduction section. 

3. Concise the method section. 

4. Bar diagram will be needed  for all the results in addition to table. 

5. Rewrite the discussion section in a sequence wise manner, remove the repetition in this section. 

6. Discuss Anti-Parkinsonian activity of Mucuna pruriens, ursolic acid and chlorogenic acid in MPTP intoxicated mouse model and relate it with your findings along with Anti-apopototic activity of Withania somnifera in Maneb and Paraquat induced mouse model. 

7. Complete editorial checking will be needed to correct the gramatical and punctuation mistakes. 

Reviewer 2 Report

This cross-sectional study presents a comprehensive clinical assessment of prodromal or early signs of Parkinson's disease in "at risk" GBA mutation carriers. The study is performed at the largest Gaucher disease clinics in the world. The selected assessments are highly relevant and  the examinations are thoroughly performed. The data are nicely presented and discussed. I have only very minor comments.

Line 139. 12-items UPSIT is often referred to as B-SIT.

Line 203: The sentence is prematurely terminated. Shall you add "are planned" at the end of the sentence?

Line 347: What kind of PET-CT? Fdopa PET? DatSCAN SPECT?

Round 2

Reviewer 1 Report

Gramatical mistakes still found in the manuscript. This should be improved by native english speaker.

Recent clinical findings should be incorporated in the discussion section of the manuscript.

I agree your paper is on clinical information, still you can correlated your findings with animal model of PD mentioned in my previous comments.

Author Response

Dear Editor, Thank you for your review. Enclosed is our response point by point, as requested: 1. Gramatical mistakes still found in the manuscript. This should be improved by native english speaker. Grammatical mistakes were corrected by three native English speakers; we have added two of the in the acknowledgment. 2. Recent clinical findings should be incorporated in the discussion section of the manuscript. We have reviewed again the literature via pubmed and have not identified new relevant findings to add. 3. I agree your paper is on clinical information, still you can correlated your findings with animal model of PD mentioned in my previous comments. We have added the following sentence to the discussion along with 2 related references (60-61), "Candidate therapeutic options include GBA-specific pharmacological chaperons and anti-inflammatory drugs such as colchicine [60] or Ursolic Acid which have demonstrated impact on neurodegeneration in various animal models including the MPTP-Induced intoxicated mouse model [61]".
